# From Sweet Corn By-Products to Carotenoid-Rich Encapsulates for Food Applications

Jelena Vulić, Vanja Šeregelj, Vesna Tumbas Šaponjac, Milica Karadžić Banjac, Strahinja Kovačević, Olja Šovljanski *, Gordana Ćetković, Jasna Čanadanović-Brunet *, Lidija Jevrić and Sanja Podunavac-Kuzmanović

Faculty of Technology Novi Sad, University of Novi Sad, Bulevar cara Lazara 1, 21000 Novi Sad, Serbia
* Correspondence: oljasovljanski@uns.ac.rs (O.Š.); jasnab@uns.ac.rs (J.Č.-B.)

**Abstract:** In the present study, carotenoids were recovered from processing sweet corn by-products (SCB). The total carotenoid content determined in the SCB extract was 1.19 mg/100 g DW, and the principal carotenoids identified by the HPLC technique were zeaxanthin, β-cryptoxanthin and lutein. Freeze- and spray-drying techniques were applied for the encapsulation of SCB extract; for this purpose, four different wall materials were used: two proteins (soy and pea) and two carbohydrates (maltodextrin and inulin). The physicochemical characteristics of eight encapsulates were determined to assess their stability. The obtained results indicate that, by using the freeze-drying method, better water activity, moisture content as well as encapsulation efficiency were achieved. Spray-drying resulted in better flowing properties. All obtained encapsulates were microbiologically safe for food applications due to the fact that the obtained results are in agreement with the requirements for consumer safety, i.e., for further food applications and scale-up processes. Chemometric classification and ranking techniques were applied to observe potential grouping among the investigated encapsulates and to select the most favorable encapsulates regarding the used wall materials and encapsulation techniques for the assessment of sustainability in food products. The most suitable wall material and encapsulation technique for the assessment of sustainability in food products was produced by freeze-drying pea protein as a wall material (FDP).

**Keywords:** sweet corn by-product; encapsulation; freeze-drying; spray-drying; chemometrics tools; ranking; classification



## 1. Introduction

One of the unavoidable characteristics of the food industry is obtaining enormous quantities of by-products, which are usually used as cheap animal feed and fertilizers. Considering that many scientific groups have reported that waste material from fruits and vegetable processing contains residues of valued substances, such as antioxidants, dietary fibers, natural colorants and aroma compounds, the main concept of the utilization of that type of waste can be directed to the formation of functional additives in food, pharmaceuticals and the cosmetic industry [1].

Corn, known as maize, originated in America and is presently one of the major food sources in the world. Besides corn grain, sweet corn represents one of the most used vegetables in North America and China, and globally, the use of this plant rapidly increases every year. The reason for the universality and acceptability of sweet corn (*Zea mays saccharata* Sturt.) lies in its sweet-tasting aroma as a result of its high sugar content, which is obtained during starch synthesis. In addition, sweet corn is classified as the most consumable vegetable in the United States [2]. The nutrition profiles of corn and sweet corn are similar, providing carotenoids, vitamins and minerals [3]. These phytochemicals promote healthy vision and antioxidants to power a baby's immune system.

To this day, many different methods for the protection of sensitive compounds from waste materials have been developed, but encapsulation within edible materials is the most promising approach of protection in the food industry, enabling longer shelf-life, controlled time or place of the release of target components as well as reductions in their degradation [4]. Moreover, by using this solution in food processing, some crucial characteristics of targeted compounds can be improved, such as solubility, bioaccessibility and bioavailability [5]. For the most used group of bioactives from fruits and vegetables, such as carotenoids, tocols and phenols, freeze-drying and spray-drying techniques are the most promising approaches for encapsulation due to simplicity, continuity, effectiveness, availability and applicability [6]. It is worth mentioning that one of the main characteristics of the freeze-drying technique is minimal thermal degradation reactions during the encapsulation process, which is very important for the protection of thermally unstable compounds such as carotenoids [6,7]. Moreover, the spray-drying technique can be a thing of choice due to its cheapness and flexibility [6].

Different chemometrics tools are widely used in all scientific fields for the explanation and interpretation of results. In the domain of food science, principal component analysis (PCA), hierarchical cluster analysis (HCA) and the sum of ranking differences (SRD) have been successfully practical for the investigation of raspberry convective drying processes [8], insect species or human consumption [9], confectionery products [10], common beans [11], different filtration experiments of rough beer [12], etc.

The investigation of the development of supplements with high contents of the antioxidants lutein and zeaxanthin have been very attractive in recent years, because the mentioned antioxidants are principally used to delay free radical accumulation and to strengthen oxidative stability. Previous data on the antioxidant activity of by-products after sweet corn processing have not been found. Hence, the present study aimed to obtain carotenoid-rich encapsulates based on sweet corn by-product extracts that are rich in these pigments, by the freeze-drying and spray-drying techniques, using four different wall materials (soy and pea proteins, maltodextrin and inulin). For the assessment of the stability of the obtained encapsulates, the physicochemical characteristics of all eight types were determined. The other aim of this study was to use classification and ranking tools in order to detect grouping among the investigated encapsulates and to detect the most suitable encapsulates for the assessment of conceivable applications as food additives.

## 2. Materials and Methods

### 2.1. Materials

Sweet corn by-product (SCB) was acquired from the "Swisslion-Takovo" industry, after producing "Juvitana" baby food. The SCB was freeze-dried and stored at freezing temperature prior to further analysis. Soy protein (S) was produced in Poland (Olimp Laboratories Sp.z oo, Dębica), maltodextrin (M) was produced in Great Britain (Battery Nutrition Limited, London), pea protein (P) was produced in Serbia (Beyond doo, Niš) and inulin (I) was produced in Serbia (Elephant Co., White City).

### 2.2. Sweet Corn By-Product Extraction

The obtaining of SCB extract was performed using the procedure explained by Šeregelj et al. [13]. The extraction of the SCB was performed 3 times by an acetone:ethanol mixture (36:64 *v/v*) in a solid to solvent ratio of 1:20 *w/v* for 10 min using a laboratory shaker (Unimax 1010, Heidolph Instruments GmbH, Kelheim, Germany) under light protection at room temperature.

### 2.3. Characterization of the Sweet Corn By-Product Extract

#### 2.3.1. Carotenoids Analysis

The total carotenoid content (TCar) in the SCB extract was tested by the spectrophotometric method used by Nagata and Yamashita [14]. The TCar was represented as mg/100 g dry weight (DW).

The HPLC analysis of carotenoids was performed using a Shimadzu Prominence chromatography system with an SPD-20AV UV-Vis detector (Shimadzu, Kyoto, Japan), following the report by Tomšik et al. [15] and the manufacturer's recommendations.

### 2.3.2. In Vitro Antioxidant Activity Analysis

The antioxidant activity was spectrophotometrically analyzed by the following methods: 2,2-diphenyl-1-picrylhydrazyl (DPPH), reducing power (RP) and a β-carotene bleaching assay (BCB), as described by Šeregelj et al. [13]. The antioxidant activity was expressed as millimoles of Trolox equivalent (TE) per 100 g of dried sample.

### 2.4. Encapsulation Process of Sweet Corn By-Product Extract

The freeze-dried and spray-dried encapsulation process, which is described by Šeregelj et al. [13], implies using four different wall materials: soy and pea proteins, maltodextrin and inulin as wall materials for the SCB extract. An amount of each (7 g) was liquefied in 10.5 mL of water at a temperature of 60 °C and was stirred until the temperature decreased to 30 °C. The same method for dissolving was applied for the mixture for spray drying, using 21 mL of water. The 40 mL of SCB extract was mixed with 1.5 mL of sunflower oil and was concentrated under reduced pressure on a rotary evaporator set at 40 °C. The mixtures were homogenized at 11,000 rpm for 3 min at room temperature and were subjected to drying.

### 2.4.1. Freeze-Drying Conditions

The previously prepared mixture was kept in a freezer during the night and was then freeze-dried at −40 °C for 2 days to ensure complete drying. The collected freeze-dried encapsulate (FD) was stored at the temperature of the freezer.

### 2.4.2. Spray-Drying Conditions

The homogenized mixture was spray-dried by a spray-drier (mini-Büchii B-190; Büchii Labortechnik, Flawil, Switzerland). The inlet temperature was 130 °C, whereas the outlet temperature was 65 ± 2 °C. The spraying air flow rate and the rate of liquid feed were 600 L/h and 8 mL/min, respectively. The spray-dried encapsulate (SD) was stored at the temperature of the freezer until further use.

### 2.5. Characterization of the Sweet Corn By-Product Encapsulates

### 2.5.1. Water Activity ($a_w$)

The water activity was analyzed by placing 3 g of encapsulates in the sample holder (LabSwiftawmeter "Novasina", Lachen, Switzerland) at 25 °C. The recording of $a_w$ values was performed after equilibration.

### 2.5.2. Moisture Content

The air oven method was used for measuring the moisture content of the encapsulates. The work temperature was 105 °C, and the measurement was performed until a constant weight was gained.

### 2.5.3. Hygroscopicity

The hygroscopicity of the samples was gained using a hermetic flask filled with a NaCl-saturated solution (75.29% RH). Each Petri dish contained 2 g of each encapsulate in a container, and after seven days, the hygroscopic moisture was evaluated and expressed as g of moisture per 100 g of dry solids.

### 2.5.4. Solubility

The applied method by Yamashita et al. [16] explains that the encapsulate sample (0.5 g) needed to be mixed in 50 mL of distilled water and stirred for half an hour at a temperature of 22 °C. The obtained mixture was centrifuged at 4000 rpm for 5 min, and

an aliquot (25 mL) of the supernatant was transferred to a pre-weight Petri dish and was dried at 105 °C for 6 h. The dried weight of the soluble solid was measured and used to calculate the percentage solubility.

2.5.5. Bulk Density (Db), Tapped Density (Dt), Carr's Index (CI) and Hausner Ratio (HR)

For the analysis of bulk density (Db), the encapsulate (10 g) was poured into a measuring cylinder, and the initial volume was noted as the bulk volume ($V_b$). The Db was calculated using Equation (1).

$$Db\ (g/mL) = \frac{m}{V_b} \tag{1}$$

For the tapped density (Dt), the encapsulate was tapped 250 times. The obtained value of the volume was written, and tapping was continued until the difference between successive volumes was less than 2%, which was registered as the tapped volume ($V_t$). The Dt was calculated using Equation (2).

$$Dt\ (g/mL) = \frac{m}{V_t} \tag{2}$$

Carr's Index (CI) and the Hausner ratio (HR) were calculated based on Equation (3) and Equation (4), respectively.

$$CI = \frac{(D_t - D_b)}{D_t} \times 100 \tag{3}$$

$$HR = \frac{D_t}{D_b} \tag{4}$$

2.5.6. Flowability and Cohesiveness

Table 1 summarizes Carr's Index (CI) and the Hausner ratio (HR), which were used for the classification of encapsulate flowability and cohesiveness. The method is described in detail by Jinapong et al. [17].

**Table 1.** Classification of encapsulate flowability and cohesiveness.

| Carr's Index | Flowability |
|:---:|:---:|
| <15 | Very good |
| 15–20 | Good |
| 20–35 | Fair |
| 35–45 | Bad |
| >45 | Very bad |
| **Hausner ratio** | **Cohesiveness** |
| <1.2 | Low |
| 1.2–1.4 | Intermediate |
| >1.4 | High |

2.5.7. Encapsulation Efficiency (EE)

The mass fractions of surface carotenoids (*SC*) and total carotenoids (*TC*) in the extracts were analyzed by the method of Barbosa et al. [18]. The same protocol was used for the encapsulation efficiency (EE) of carotenoids. The carotenoid quantification was performed following the previously mentioned spectrophotometric procedure. The encapsulation efficiency was calculated by using Equation (5). The control sample was the wall materials without the extracts.

$$EE\ (\%) = \frac{TC - SC}{TC} \times 100 \tag{5}$$

### 2.5.8. Color Parameters

The color measurements were performed by a Minolta reflectance colorimeter (Minolta Chroma Meter CR-300, Minolta, Osaka, Japan). Equation (6) was used for the calculation of the Chroma or saturation (C*).

$$C* = \sqrt{a^2 + b^2} \qquad (6)$$

### 2.6. Microbiological Profile

The microbiological profile of the raw materials and the obtained encapsulated samples were tested in order to determine the safety of using the selected materials, hygiene during the encapsulation process as well as the safety and stability of the final products. All analyses were conducted following standardized methods used in the laboratory for food microbiology: ISO 4833-1:2013 (aerobic mesophilic bacteria) [19], ISO 21527-2:2008 (yeast and moulds) [20], ISO 21528-2:2017 (Enterobacteriaceae) [21] and ISO/DIS 6888-1:2018 (*Staphylococcus aureus*) [22]. All analyses were performed in triplicate.

### 2.7. Chemometric Analysis

Principal component analysis was used for the estimation of an uncorrelated set of variables, which were ordered such that the first few retained most of the variation present in all of the original variables. New (latent) variables are calculated using a combination of the original variables [23]. For distributing a group of objects into clusters (determination of the distance between two samples), hierarchical cluster analysis (HCA) was conducted using the method described by Miller and Miller [23]. The obtained results are presented as a dendrograms. The sum of ranking differences (SRD) analysis was introduced by Héberger [24] and Héberger and Kollár-Hunek [25], which offers an evaluation of differences between ideal and actual rankings. The main rule includes the idea that, as the SRD value comes closer to zero, the model becomes better [25].

### 2.8. Statistical Analysis

All analyses were conducted in triplicate, and the obtained results are denoted as the means $\pm$ standard deviation. The origin 8.0 SRO software package and Microsoft Office Excel 2010 software were used for mathematical calculations. Significant differences were calculated using an ANOVA ($p < 0.05$). PCA and HCA were performed using Statisticasoftware (version 10.0, Washington, D.C., USA).

## 3. Results and Discussion

### 3.1. Characterization of the Sweet Corn By-Product Extract

It is believed that carotenoids contribute to the yellow color of sweet corn. The TCar found in the extract of sweet corn by-product, which remains after this cultivar processing, was 1.19 mg/100 g. The principal carotenoids in SCB extract identified by the HPLC technique were zeaxanthin (1.15 mg/100 g), β-cryptoxanthin (0.34 mg/100 g) and lutein (0.03 mg/100 g). Song et al. [26] reported the carotenoid profiles of different corn genotypes and concluded that every sweet corn type at each stage has lutein, zeaxanthin and all-trans-α-cryptoxanthin as the principal shape of the mentioned pigment. The flesh color of sweet corn can vary due to the fact that the cultivation of it is affected by variety, maturity, post-harvest storage conditions and season. In the past decades, the development of high-carotenoid corn varieties rich in lutein or zeaxanthin have had priority due to the fact that these two shapes of carotenoids enable free radical accumulation delay and fortify oxidative stability [27]. Furthermore, using antioxidants is a good way to prevent many modern diseases [28], and previous data on the antioxidant activity of by-products after sweet corn processing were not found.

In this investigation, the free radical scavenging capacity of SCBE was analyzed by three assays: DPPH, RP and BCB (Table 2). For the analysis of free radical scavenging activity, the stable DPPH was selected, and reducing the power of bioactive compounds was measured due to the fact that is associated with their ability to transfer electrons. The

last one, the BCB assay, offers data about lipid peroxidation in oil-in-water emulsions, measuring the loss of the yellow color of β-carotene in reactions with formed linoleic acid radicals, which can be minimized if antioxidants are present.

**Table 2.** Carotenoid content and antioxidant activity of SCB extract.

| Total Carotenoids Content (mg/100 g DW) | |
|---|---|
| TCar | $1.19 \pm 0.07$ |
| **Individual Carotenoids Content (mg/100 g DW)** | |
| Lutein | $0.03 \pm 0.01$ |
| Zeaxanthin | $1.15 \pm 0.04$ |
| β-cryptoxanthin | $0.34 \pm 0.01$ |
| **Antioxidant Activity (µmol TE/100 g DW)** | |
| DPPH | $633.11 \pm 37.73$ |
| RP | $144.87 \pm 1.80$ |
| BCB | $73.54 \pm 6.59$ |

The results are presented as the mean value $\pm$ standard deviation ($n = 3$).

### 3.2. Characterization of the Sweet Corn By-Product Encapsulates

According to Šeregelj et al. [29], the drying parameters and the characteristics of the wall material are the main factors that can affect encapsulation efficiency. Hence, spray-drying and freeze-drying as encapsulation techniques and four different wall materials (soy and pea proteins, maltodextrin and inulin) for the encapsulation of SCBE were used in this study. The physicochemical characteristics of the obtained eight encapsulates is shown in Table 3.

**Table 3.** Physicochemical properties of sweet corn by-product encapsulates.

| Characteristics | FDS | FDP | FDM | FDI | SDS | SDP | SDM | SDI |
|---|---|---|---|---|---|---|---|---|
| Water activity (a$_w$) | $0.02 \pm 0.00$ [a] | $0.02 \pm 0.00$ [a] | $0.02 \pm 0.00$ [a] | $0.02 \pm 0.00$ [a] | $0.11 \pm 0.00$ [d] | $0.22 \pm 0.01$ [e] | $0.05 \pm 0.00$ [b] | $0.08 \pm 0.00$ [c] |
| Moisture content (g/100 g) | $0.76 \pm 0.00$ [a] | $1.21 \pm 0.01$ [d] | $0.87 \pm 0.02$ [b] | $0.97 \pm 0.01$ [c] | $2.61 \pm 0.03$ [f] | $4.80 \pm 0.03$ [h] | $3.23 \pm 0.05$ [g] | $2.32 \pm 0.01$ [e] |
| Higroscopicity (g/100 g) | $12.92 \pm 0.05$ [c] | $12.68 \pm 0.03$ [b] | $14.96 \pm 1.12$ [e] | $14.99 \pm 1.26$ [e] | $13.46 \pm 0.98$ [d] | $13.50 \pm 0.7$ [d] | $8.06 \pm 0.17$ [a] | $13.63 \pm 2.33$ [d] |
| Solubility (g/100 g) | $20.10 \pm 0.45$ [c] | $12.80 \pm 1.10$ [b] | $91.50 \pm 2.65$ [g] | $87.81 \pm 0.98$ [f,g] | $20.70 \pm 2.15$ [c] | $2.50 \pm 0.08$ [a] | $83.71 \pm 4.14$ [e] | $63.98 \pm 3.24$ [d] |
| Bulk density; Db (g/mL) | $0.60 \pm 0.00$ [a] | $0.81 \pm 0.01$ [b] | $0.81 \pm 0.02$ [b] | $0.86 \pm 0.02$ [c] | $2.08 \pm 0.02$ [e] | $2.27 \pm 0.05$ [f] | $1.78 \pm 0.12$ [d] | $2.27 \pm 0.04$ [f] |
| Tapped density; Dt (g/mL) | $1.04 \pm 0.00$ [a] | $1.79 \pm 0.00$ [d] | $1.14 \pm 0.03$ [b] | $1.25 \pm 0.02$ [c] | $2.50 \pm 0.01$ [f] | $2.27 \pm 0.02$ [e] | $2.50 \pm 0.01$ [f] | $2.50 \pm 0.03$ [f] |
| Carr's Index (CI) | $42.31 \pm 0.00$ [e] | $54.75 \pm 0.0$ [f] | $28.95 \pm 0.05$ [c] | $31.2 \pm 0.04$ [d] | $16.80 \pm 0.03$ [b] | 0 | $28.80 \pm 0.13$ [c] | $9.20 \pm 0.07$ [a] |
| Hausner ratio (HR) | $1.73 \pm 0.00$ [d] | $2.21 \pm 0.0$ [e] | $1.41 \pm 0.05$ [c] | $1.45 \pm 0.04$ [c] | $1.20 \pm 0.03$ [b] | 0 | $1.40 \pm 0.13$ [c] | $1.10 \pm 0.07$ [a] |
| Flowability | Bad | Very bad | Fair | Fair | Good | Very good | Fair | Good |
| Cohesiveness | High | High | High | High | Intermediate | Low | Intermediate | Low |
| Encapsulation efficiency; EE (%) | $39.74 \pm 2.14$ [b,c] | $92.74 \pm 5.33$ [f] | $55.47 \pm 2.12$ [e] | $40.03 \pm 2.24$ [c] | $51.66 \pm 2.30$ [d] | $81.69 \pm 3.47$ [e] | $36.51 \pm 4.65$ [b] | $25.89 \pm 1.21$ [a] |
| **CIE Lab** | | | | | | | | |
| L* | $75.68 \pm 0.02$ [d] | $71.51 \pm 0.00$ [a] | $86.04 \pm 0.01$ [e] | $93.74 \pm 0.01$ [g] | $72.88 \pm 0.01$ [b] | $75.29 \pm 0.01$ [c] | $92.98 \pm 0.01$ [f] | $93.97 \pm 0.01$ [h] |
| A* | $-0.04 \pm 0.01$ [f] | $2.26 \pm 0.03$ [h] | $-4.56 \pm 0.01$ [a] | $-2.74 \pm 0.00$ [d] | $-0.11 \pm 0.01$ [e] | $1.40 \pm 0.02$ [g] | $-2.89 \pm 0.01$ [c] | $-2.60 \pm 0.01$ [b] |
| B* | $19.77 \pm 0.01$ [d] | $22.20 \pm 0.01$ [f] | $22.39 \pm 0.01$ [g] | $10.82 \pm 0.01$ [b] | $22.98 \pm 0.01$ [h] | $21.33 \pm 0.01$ [e] | $12.38 \pm 0.01$ [c] | $10.17 \pm 0.01$ [a] |
| C* | $19.77 \pm 0.04$ [d] | $22.31 \pm 0.04$ [f] | $22.85 \pm 0.03$ [g] | $11.16 \pm 0.02$ [b] | $22.98 \pm 0.03$ [h] | $21.38 \pm 0.04$ [e] | $12.71 \pm 0.03$ [c] | $10.50 \pm 0.03$ [a] |

Values sharing the same letter in the same raw are not significantly different at the 0.05 level.

The free water and availability for potential microbial contamination can decrease the quality of food and food-related sources. Therefore, the a$_w$ and moisture content are crucial characteristics because of their role in lipid peroxidation, microbial growth and enzymatic and non-enzymatic reactions in food. Higher amounts of water activity indicate increased free water available for biochemical reactions, which affect the products' shelf life. It is believed that a$_w$ values below 0.6 can inhibit microbial growth. According to the obtained results, the a$_w$ values of SCEs indicate the range of microbiologically stable environments (FDS = 0.02, and SDP = 0.22). A generally higher a$_w$ value of SDS compared to FDS can be observed, which is comparable with results from previous research [30,31]. Similarly, a

lower moisture level indicates a higher rate of preservation and stability of the encapsulates. The lowest moisture content was generally noticed after the freeze-drying process, where soy encapsulates showed the lowest content (0.76 g/100 g) (Table 3).

Hygroscopicity, the ability to absorb moisture from a relatively humidity environment [32], is crucial for food technology for encapsulates. Defining this parameter offers the best possibility to maintain the best storage conditions. After one week, the highest hygroscopicity was reached in FDI (14.99 g/100 g), compared to the lowest SDM (8.06 g/100 g). Tonon et al. [33] reported that the encapsulates with low moisture have the greatest capacity to absorb water from the environment. On the other side, Ahmed et al. [34] found that the hygroscopicity of encapsulates was affected by wall material, without a direct association to moisture content. Our findings are in agreement with the conclusion of Ahmed et al. [34] since inulin influenced the hygroscopicity of the encapsulates. The explanation for these properties is the high water affinity of inulin, which has hydrophilic groups [35]. The same explanation can be related to the results of the solubility or the ability of encapsulates to form a solution in water, where carbohydrate wall materials showed better solubility for both drying methods.

The SD encapsulates (0.45 and 0.77 g/mL) had higher values of bulk and tapped density, which correspond to the gained smaller particle size. Correa-Filho et al. [36] also emphasized that the bulk density of spray-dried encapsulates is affected by the mentioned characteristics. Furthermore, the high bulk and tapped density of encapsulates enables packaging in smaller containers and decreases the possibility for powder oxidation.

Using Carr's index as an indicator for flowability and the Hausner ratio as a measurment of cohesiveness [17], the encapsulates' flowability and cohesiveness showed that freeze-dried samples had fair, bad and very bad flowability and high cohesiveness, whereas spray-dried encapsulates had fair, good and very good flowability with low and intermediate cohesiveness. The flow property is dependent on the wall material characteristics as well as the technique applied for encapsulation. Zhang et al. [37] highlighted that encapsulates with good flowing assets are suitable for handling and processing operations.

### 3.3. Microbiological Profile

Regardless of the fact that they are in a dry, powdery state, many raw materials represent a suitable and nutrient-rich environment for the growth of a wide range of microorganisms [4]. Additionally, fruit and vegetable waste contains a high level of moisture and nutrients, and it can be stored inappropriately after food processing, therefore being a source of microbiological contamination [38]. Due to these facts, it was necessary to analyze the microbiological profile of all initial materials for encapsulate production. As shown in Table 4, sweet corn by-product (SCB) did not initially contain a large number of microorganisms. The numbers of aerobic and mesophilic bacteria, Enterobacteriaceae and S. aureus were below 1 log CFU/mL, and yeasts and molds were detected in higher concentrations, i.e., 1.05 ± 0.04 log CFU/mL. For all raw materials used as coating agents (soy protein, pea protein, maltodextrin and inulin), it can be observed as having similar responses to hygienic parameters, i.e., Enterobacteriaceae. *S. aureus* was absent, and different concentrations of basic parameters were detected (Table 4). Although the presence of microbiological contamination was confirmed, the obtained concentration was in the range of acceptability according to the legal framework for food microbiological safety (Guide to microbiological criteria for food, 2011). The obtained extract and encapsulated samples were microbiologically safe for food applications due to the fact that the obtained results were in agreement with the requirements for consumers' safety. All tested microbiological parameters were below the detection limit, which suggests that encapsulation process was hygienically adequate. Finally, it can be concluded that the utilized processes, equipment as well as raw materials were minimally contaminated and were therefore suitable for further food applications and scale-up processes.

**Table 4.** Microbiological profiles of raw materials and encapsulated products.

| Sample | Number of Targeted Microorganisms (log CFU */g) | | | |
|---|---|---|---|---|
| | Aerobic and Mesophilic Bacteria | Yeasts and Molds | Enterobacteriaceae | *Staphylococcus aureus* |
| SCB | <1 | 1.05 ± 0.04 | <1 | <1 |
| soy protein | 1.4 ± 0.56 | 2.04 ± 0.41 | <1 | <1 |
| pea protein | <1 | 1.2 ± 0.03 | <1 | <1 |
| maltodextrin | 1.2 ± 0.00 | 1.2 ± 0.56 | <1 | <1 |
| inulin | <1 | 1.05 ± 0.01 | <1 | <1 |
| freeze-dried extract | <1 | <1 | <1 | <1 |
| spray-dried extract | <1 | <1 | <1 | <1 |
| FDS | <1 | <1 | <1 | <1 |
| FDG | <1 | <1 | <1 | <1 |
| FDM | <1 | <1 | <1 | <1 |
| FDI | <1 | <1 | <1 | <1 |
| SDS | <1 | <1 | <1 | <1 |
| SDG | <1 | <1 | <1 | <1 |
| SDM | <1 | <1 | <1 | <1 |
| SDI | <1 | <1 | <1 | <1 |

* CFU—colony forming unit.

### 3.4. Chemometric Analysis

PCA analysis was applied to determine potential similarities and dissimilarities among the investigated encapsulates and to observe potential grouping among them according to similarities in the space of experimentally determined physico-chemical characteristics. The results of PCA analysis for the physico-chemical characteristics of eight investigated encapsulates obtained by encapsulating the extract of sweet corn by-product resulted in a model with seven principal components. The eigenvalues for principal components, total variance percent and their cumulative eigenvalues are presented in Supplementary Data Table S1. For further analysis, first, three principal components were taken into account. These three principal components covered 89.30% of the total variance, with PC1 contributing 50.39%, PC2 contributing 24.78% and PC3 contributing 14.13%. Loadings and scores plots for PC1–PC2 are presented in Figure 1 and for PC1–PC3 in Figure 2.

From the loadings plot (Figure 1A) it can be seen that, at the position of the eight examined encapsulates on the scores plot (Figure 1B), there was almost an equal influence along the PC1 axis for bulk density, moisture content and water activity. Along the PC2 axis, the most dominant influence is the encapsulation efficiency and chroma. Along the PC1 axis, bulk density, moisture content and water activity have a positive coefficient of latent variables. In relation to the PC2 axis, the encapsulation efficiency and chroma also have a positive coefficient of latent variables. From the scores plot, it can be concluded that the examined encapsulates are grouped in relation to the way they are produced along the PC1 axis. At the positive end of the PC1 axis, positioned encapsulations are produced by spray-drying (SDS, SDP, SDM and SDI), i.e., with those with higher values of bulk density, moisture content and water activity. Those encapsulates were produced by freeze-drying (FDS, FDP, FDM and FDI), i.e., with those with lower values of bulk density, moisture content and water activity, positioned on the negative part of the PC1 axis.

The analysis of the principal components for PC1–PC3 is shown in Figure 2. From the loadings plot (Figure 2A), it can be seen that, at the position of the eight examined encapsulates on the scores plot (Figure 2B), the same influence along the PC1 axis has the same physico-chemical characteristics (bulk density, moisture content and water activity) as for the PC1–PC2 display. Along the PC2 axis, the most dominant influence has hygroscopicity. Along the PC1 axis, bulk density, moisture content and water activity have a positive coefficient of latent variables, and along the PC2 axis, hygroscopicity has a negative coefficient of latent variables. Thus, the grouping of the examined encapsulates is very similar to that for PC1–PC2 along the PC1 axis. From the scores plot, it can be seen

that, at the positive end of the PC1 axis, there are encapsulates produced by spray-drying (SDS, SDP, SDM and SDI), i.e., with those with higher values of bulk density, moisture content and water activity. On the negative end of the PC1 axis, encapsulates produced by freeze-drying (FDS, FDP, FDM and FDI) are positioned, i.e., those with lower values of bulk density, moisture content and water activity.

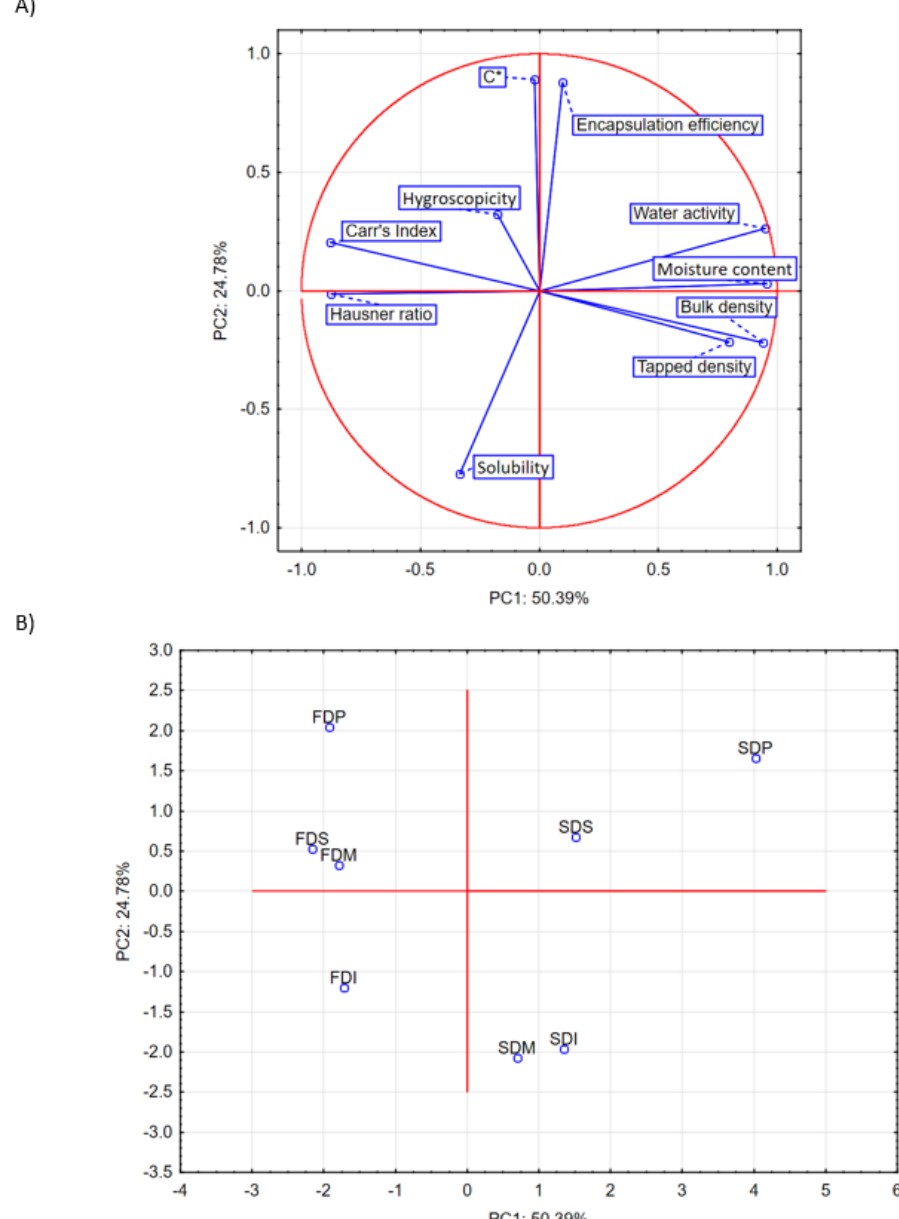

**Figure 1.** PCA graph for physico-chemical characteristics of eight investigated encapsulates: (**A**) loadings and (**B**) scores plot.

Hierarchical cluster analysis (HCA) was conducted on the same data set, and the results of the HCA analysis are shown through a dendrogram, indicating a slightly different grouping as that in the PCA analysis. The dendrograms obtained using Ward´s method and Euclidean distances are presented in Figure 3. The dendrogram is divided into two clusters, both with two subclusters. The first cluster contains encapsulates produced on the soy and pea proteins as the wall materials with both freeze-drying and spay-drying (FDS, FDP, SDS and SDP), and the second one contains encapsulates produced on the maltodextrin and inulin as the wall materials (FDM, FDI, SDM and SDI).

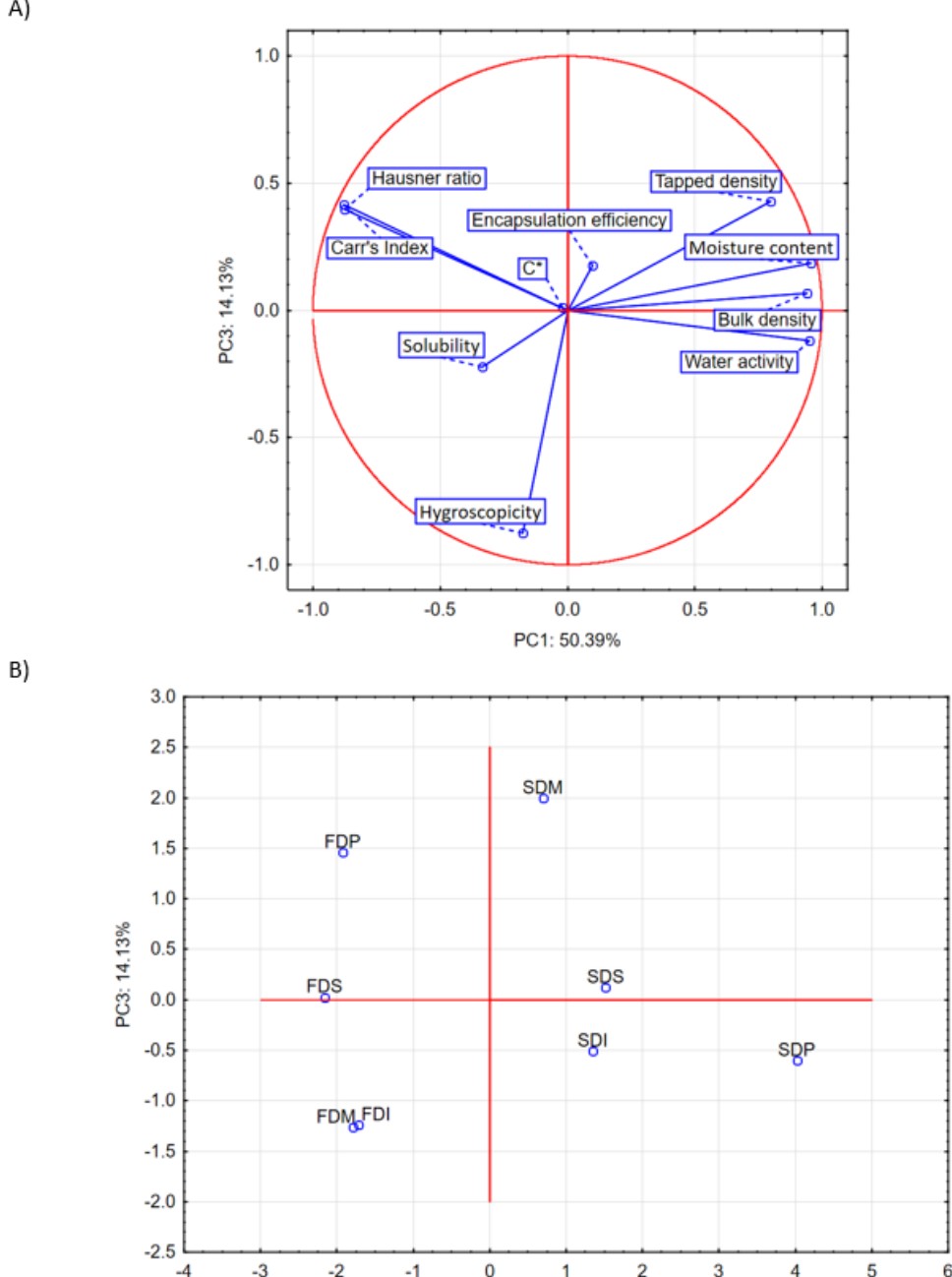

**Figure 2.** PCA analysis results for PC1–PC3 for physico-chemical characteristics of eight investigated encapsulates: (**A**) loadings and (**B**) scores plot.

The SRD analysis is based on the golden standard, or reference ranking, in relation to which the ranking of investigated encapsulates is performed. This type of analysis was performed to select encapsulates with the most favorable physico-chemical characteristics along with the maximal encapsulation efficiency. Before performing the analysis, the physico-chemical characteristics were organized in the form of a matrix, in which the values of the experimentally obtained physico-chemical characteristics were placed in rows, and the encapsulates were placed in columns. The maximum values of solubility, bulk density, tapped density, encapsulation efficiency and chroma, as well as minimum values of water activity, moisture content, hygroscopicity, Carr's Index and Hausner ratio, were taken as the reference rankings. As the SRD value comes closer to zero, the encapsulate becomes better.

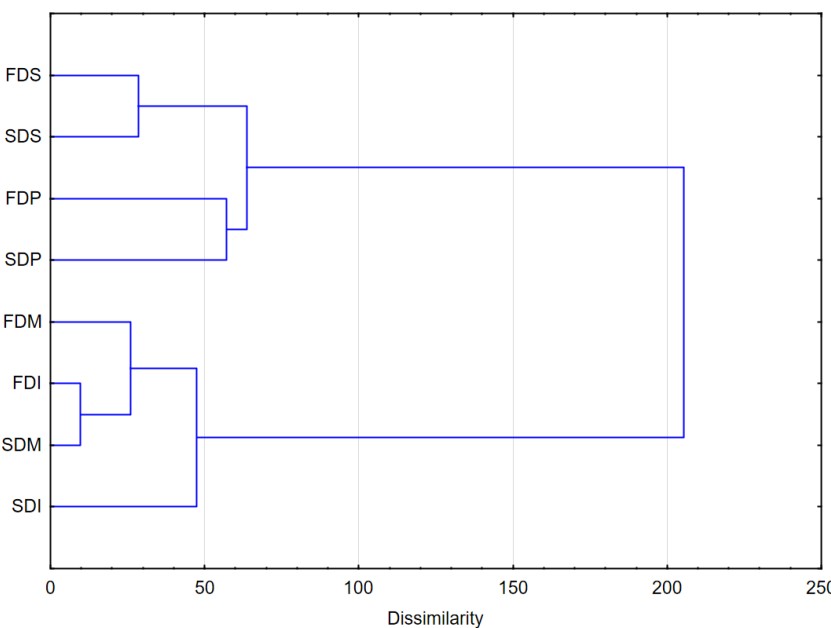

**Figure 3.** HCA analysis results shown as a dendrogram of eight examined encapsulates in the space of determined physico-chemical characteristics.

The results shown in Figure 4 suggest that the examined encapsulates were grouped into seven groups, based on their SRD values. The most suitable wall material and encapsulation technique for the assessment of sustainability in food products was produced by freeze-drying pea protein as a wall material (FDP). The FDP encapsulate represents a compromise regarding its closest position of the reference rank, as it has the highest value of encapsulation efficiency and also possesses other favorable physico-chemical characteristics. Immediately after FDP, the SDS (produced using spray-drying on soy protein as a wall material) encapsulate is positioned. The most unfavorable are encapsulates produced by spray-drying on pea protein (SDP) and inulin (SDI) as wall materials, which are located farthest from the reference rank, which implies that these encapsulates should not be taken into account in future analyses.

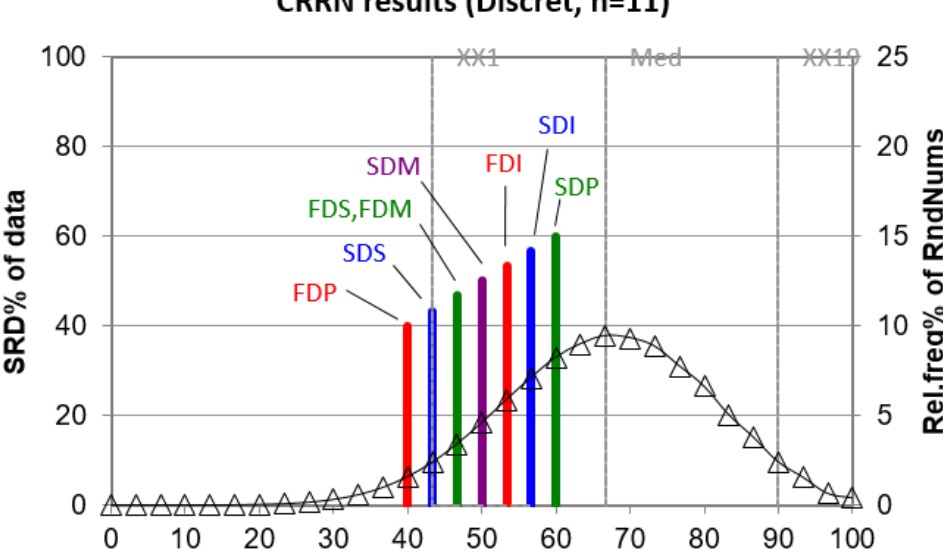

**Figure 4.** SRD analysis results for physico-chemical characteristics of eight investigated encapsulates.

## 4. Conclusions

The SCB extract was efficaciously encapsulated by freeze- and spray-drying techniques using four wall materials. The different drying techniques, as well as the used wall materials, were strongly affected by the SCB encapsulate properties. The applied PCA and HCA analyses enabled grouping and emphasizing the main variances between the eight investigated encapsulates. The used SRD analysis indicated which encapsulates should be avoided in future analyses and those which are the most suitable for the assessment of sustainability in food products. Thus, the most suitable wall material and encapsulation technique for the assessment of sustainability in food products was produced by freeze-drying on pea protein as a carrier (FDP), and the most unfavorable were encapsulates produced by spray-drying on pea protein (SDP) and inulin (SDI) as wall materials. The microbiological results suggest that all samples were safe for food applications. All tested microbiological parameters were below the detection limit, which suggests that the encapsulation process was hygienically adequate. Finally, it can be concluded that the utilized processes, equipment as well as raw materials were minimally contaminated and therefore suitable for further food applications and scale-up processes.

**Supplementary Materials:** The following supporting information can be downloaded at: https://www.mdpi.com/article/10.3390/pr10081616/s1, Table S1: Total variance explained via eigenvalues.

**Author Contributions:** Conceptualization, V.Š. and J.V.; methodology, S.K., G.Ć. and L.J.; validation, V.Š., O.Š. and M.K.B.; formal analysis, V.Š., O.Š. and J.V.; investigation, J.V., J.Č.-B and S.P.-K.; data curation, M.K.B. and S.K.; writing—original draft preparation, V.Š., O.Š. and M.K.B.; writing—review and editing, V.Š.; visualization, M.K.B.; supervision, J.Č.-B., V.T.Š., G.Ć., S.P.-K. and L.J. All authors have read and agreed to the published version of the manuscript.

**Funding:** This research received no external funding.

**Institutional Review Board Statement:** Not applicable.

**Informed Consent Statement:** Not applicable.

**Data Availability Statement:** Not applicable.

**Acknowledgments:** The support of the Ministry of Education, Science and Technological Development of the Republic of Serbia (contract no. 451-03-68/2022-14/200134) is gratefully acknowledged.

**Conflicts of Interest:** The authors declare no conflict of interest.

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
