# Peer review of "From Sweet Corn By-Products to Carotenoid-Rich Encapsulates for Food Applications"

_processes, doi:10.3390/pr10081616_

Round 1

Reviewer 1 Report

The authors conducted a study to evaluate the effects of two drying methods and four encapsulating materials (two proteins and two carbohydrates), for a total of eight distinct combinations, on the encapsulation of sweet corn processing by-products. The authors performed several physico-chemical analysis and evaluated the results applying different chemometrics and ranking techniques. In the conclusions the authors suggested that some of the alternatives studied in the present work were not sufficiently satisfactory and suggested that the best encapsulation technique would be freeze-drying with pea protein as the wall material.

The work is well done. The topic is properly and clearly introduced. The methods and the results are well subdivided and described. I appreciated the effort to provide a comparison with the existing literature for each individual result obtained in addition to the numerical data.

I recommend the present paper to be accepted in the present form.

Author Response

Reviewer 1

Dear Reviewer 1,

We would like to thank you for the time, effort and expertise that you contributed to reviewing our manuscript “From sweet corn by-product to carotenoid-rich encapsulates for food application“.

Reviewer 1

The authors conducted a study to evaluate the effects of two drying methods and four encapsulating materials (two proteins and two carbohydrates), for a total of eight distinct combinations, on the encapsulation of sweet corn processing by-products. The authors performed several physico-chemical analysis and evaluated the results applying different chemometrics and ranking techniques. In the conclusions the authors suggested that some of the alternatives studied in the present work were not sufficiently satisfactory and suggested that the best encapsulation technique would be freeze-drying with pea protein as the wall material.

The work is well done. The topic is properly and clearly introduced. The methods and the results are well subdivided and described. I appreciated the effort to provide a comparison with the existing literature for each individual result obtained in addition to the numerical data.

I recommend the present paper to be accepted in the present form

AUTHORS: Thank you for all comments and valuable review. We hope that our revised version of the manuscript will be accepted for publication in “Processess“

Reviewer 2 Report

1.      Provide the obtained results in the abstract. The current form seems to be like introduction.

2.      Quality of images must be improved. For example, use some plotting software like origin and provide the quality images.

3.      Overall, the organization and interpretation of result is poor. It should be improved.

4.      Novelty of the work should be highlighted in the introduction part.

5.      Some statement should be revised with proper citation; Journal of Hazardous Materials, 412, 125245 (2021); Nanotechnology 30 (2019) 39200

Author Response

Reviewer 2

Dear Reviewer 2,

We would like to thank you for the time, effort and expertise that you contributed to reviewing our manuscript “From sweet corn by-product to carotenoid-rich encapsulates for food application“.

We will give your comments step by step through this letter. We have carefully responded to each of the referees’ comments in the text below and indicated where we have revised the manuscript. We hope that our revised version of the manuscript will be accepted for publication in “Processes“.

Reviewer 2

  1. Provide the obtained results in the abstract. The current form seems to be like introduction.

AUTHORS: Thank you. We add some comments about the main results in abstract which are related to differences between encapsulates obtained by freeze- and spray-drying techniques.

  1. Quality of images must be improved. For example, use some plotting software like origin and provide the quality images.

AUTHORS: Thank you for your suggestion. We improve the images quality.

  1. Overall, the organization and interpretation of result is poor. It should be improved.

AUTHORS: Thank you very much for your suggestion. We would like to inform you that we used standard tools for the presentation of the results, like Origin 8.0 SRO software package and Microsoft Office Excel 2010 software for mathematical calculation. Significant differences were calculated by ANOVA (p < 0.05). PCA and HCA were performed using Statistica v 10.0 software. We also presented the physicochemical characteristics of encapsulates in Tables, for easier comparison, given that we have eight samples.

  1. Novelty of the work should be highlighted in the introduction part.

AUTHORS: Thank you. We add the novelty of the work in the introduction part.

  1. Some statement should be revised with proper citation; Journal of Hazardous Materials, 412, 125245 (2021); Nanotechnology 30 (2019) 39200

AUTHORS: Thank you for your help. Although these papers that you proposed are really good, it is not related to our manuscript. Sorry for the inconvenience this may cause.

Reviewer 3 Report

In my opinion, the article is correctly prepared both from the substantive and editorial point of view. The aim of the paper is correctly formulated. The authors have clearly and precisely planned the research experiment and selected the research methods. 

Author Response

Reviewer 3

Dear Reviewer 3,

We would like to thank you for the time, effort and expertise that you contributed to reviewing our manuscript “From sweet corn by-product to carotenoid-rich encapsulates for food application“.

Reviewer 3

In my opinion, the article is correctly prepared both from the substantive and editorial point of view. The aim of the paper is correctly formulated. The authors have clearly and precisely planned the research experiment and selected the research methods.

AUTHORS: Thank you for your valuable review of our work. We hope that our revised version of the manuscript will be accepted for publication in “Processess“.
